# Prevalence, Microbiological Profile, and Risk Factors of Healthcare-Associated Infections in Intensive Care Units: A Retrospective Study in Aljouf, Saudi Arabia

**DOI:** 10.3390/microorganisms13081916

**Published:** 2025-08-17

**Authors:** Issra Taresh Alshammari, Yasir Alruwaili

**Affiliations:** 1Department of Clinical Laboratory Sciences, College of Applied Medical Sciences, Jouf University, Sakaka 72388, Saudi Arabia; essrataresh@gmail.com; 2Center for Health Research and Innovations, Deanship of Graduate Studies and Scientific Research, Jouf University, Sakaka 72388, Saudi Arabia

**Keywords:** healthcare-associated infections (HAIs), intensive care unit (ICU), multidrug-resistant organisms (MDROs), invasive devices, infection control, *Klebsiella pneumoniae*, *Acinetobacter baumannii*, Saudi Arabia, Aljouf

## Abstract

Hospital infection prevention is critical to patient safety, yet data on the prevalence and contributing factors of healthcare-associated infections (HAIs) in Aljouf, Saudi Arabia, are scarce. This retrospective cross-sectional study aimed to investigate the prevalence, microbiological profile, and associated risk factors of HAIs among intensive care unit (ICU) patients in a referral hospital between January 2020 and December 2023. Medical records of 260 ICU patients were reviewed for demographic details, comorbidities, infection types, pathogens, and invasive device use. Forty patients (15.38%) developed HAIs with the highest prevalence in 2020 (50.0%). Infections were more common in males (56.5%) and those aged ≥56 years (54.6%). The predominant infections were catheter-associated urinary tract infections (47.5%), ventilator-associated pneumonia (35.0%), and central line-associated bloodstream infections (17.5%). *Klebsiella pneumoniae* (35.0%) and *Acinetobacter baumannii* (27.5%), pathogens commonly associated with multidrug resistance, were the most frequently isolated organisms. All HAI cases involved invasive device use with the use of three or more devices significantly increasing infection risk (*p* < 0.05). Additionally, 85% of infected patients had chronic conditions, primarily hypertension or diabetes. These findings emphasize the urgent need for strengthened infection control practices and targeted antimicrobial strategies to reduce HAIs and improve ICU patient outcomes in underreported regions.

## 1. Introduction

Healthcare-associated infections (HAIs) are a major global health concern that contribute significantly to patient morbidity and mortality, prolonged hospital stays, and increased healthcare costs. They are defined by the Centers for Disease Control and Prevention/National Healthcare Safety Network (CDC/NHSN) as infections occurring ≥48 h after hospital admission, excluding those present or incubating at the time of admission [1]. The World Health Organization (WHO) estimates that 7% of hospitalized patients in high-income countries and up to 10% in low- and middle-income countries will acquire at least one HAI during their stay [2]. This risk is even higher in patients admitted to intensive care units (ICUs), who are particularly vulnerable owing to their critical condition and exposure to invasive medical devices such as ventilators and central lines [3].

Healthcare-associated infections in ICUs most commonly manifest as ventilator-associated pneumonia (VAP), catheter-associated urinary tract infections (CAUTIs), central line-associated bloodstream infections (CLABSIs), and surgical site infections (SSIs), many of which are caused by multidrug-resistant organisms (MDROs) [4]. These infections pose substantial challenges to infection prevention and patient management, especially in resource-constrained settings. For example, VAP results from bacterial colonization of the lower respiratory tract in mechanically ventilated patients, while CLABSIs involves the direct introduction of pathogens into the bloodstream through central venous catheters [5]. Infections caused by resistant organisms, such as *Klebsiella pneumoniae*, *Candida auris*, and *Clostridioides difficile*, further complicate treatment and are associated with higher mortality rates [6]. A major cross-continental study conducted by the International Nosocomial Infection Control Consortium (INICC) assessed the worldwide prevalence of HAIs. Data collected from 2015 to 2020 across 630 ICUs in 45 countries revealed 4270 CLABSI incidents, 7635 VAP cases, and 3005 CAUTI cases. The combined device-associated HAI (DA-HAI) rate was 7.28% with an incidence of 10.07 DA-HAIs per 1000 patient-days [7]. These findings emphasize the considerable global burden of HAIs affecting millions of patients annually.

During the coronavirus disease 2019 (COVID-19) pandemic, several healthcare systems showed increasing HAI incidence, which was attributed to increased patient loads, staff shortages, and the limited availability of personal protective equipment [8]. These pressures highlight the fragility of infection control practices in ICUs and underscore the importance of ongoing surveillance.

In Saudi Arabia, studies on HAIs have grown in recent years but remain limited in scope and geography. A national surveillance study conducted across 106 hospitals showed HAI rates of 2.57 CLABSIs, 1.08 CAUTIs, and 4.21 ventilator-associated events per 1000 device-days [9]. However, most of these studies were conducted in large tertiary hospitals in urban centers with minimal data from peripheral or northern regions. At the regional level, a study conducted in Hail to analyze 591 suspected HAI cases in a tertiary hospital ICU identified 163 bacterial isolates, including 70 VAP cases, 39 CAUTIs, and 25 CLABSIs [10]. Similarly, a study conducted in Arar revealed a 7.8% HAI incidence rate in maternity and pediatric hospitals with UTIs accounting for nearly half of all cases [11]. Despite these findings, other northern regions such as Aljouf remain underrepresented in national HAI data. Locally, a study conducted in a general hospital in Aljouf between 2020 and 2022 showed a gradual decrease in SSI cases, which was potentially due to strict infection control measures and active follow-up in the region. The culture of the specimens revealed methicillin-resistant *Staphylococcus aureus* and *K. pneumoniae* as the dominant isolates [12].

These findings highlight the importance of infection prevention efforts; however, comprehensive data on all HAI types and the effectiveness of current prevention strategies in Aljouf remain unavailable. Given the vulnerability of ICU patients and the critical need for regional surveillance, the aim of this study was to investigate the prevalence, microbiological characteristics, and risk factors of HAIs among ICU patients in Aljouf, thereby supporting evidence-based infection prevention strategies and strengthening regional healthcare outcomes.

## 2. Materials and Methods

### 2.1. Study Design and Population

This retrospective cross-sectional study was conducted in Aljouf, northern Saudi Arabia, covering the period from January 2020 to December 2023. Aljouf comprises 3 main administrative areas: Sakaka, Qurayyat, and Dumat Al-Jandal. The region’s healthcare infrastructure includes 2 specialty hospitals in Sakaka, each with a 300-bed capacity and functioning as a referral center, along with general hospitals in Qurayyat (260 beds) and Dumat Al-Jandal (130 beds). Collectively, these facilities serve a population of 595,822 residents, according to the 2022 national census [13].

This study included all ICU patients who developed healthcare-associated infections (HAIs) after ICU admission during their stay at a randomly selected hospital in the region within the study period. Patients with incomplete medical records, HAIs documented before ICU admission, or infections present or incubating at the time of ICU admission were excluded to ensure data accuracy.

For age-group analysis, pediatric patients (0 ≤ 18 years) were combined with young adults (18–25 years) into a single ≤ 25 years category primarily because younger pediatric ICU patients are routinely referred to the Maternity and Children’s Hospital, resulting in very few pediatric cases in the study hospital. Consequently, the number of patients under 18 was small (*n* = 13), more than half of whom were over 14 years old, and only one developed an HAI—insufficient to provide a representative picture of pediatric ICU epidemiology in our setting.

Ethical approval was obtained from the Institutional Review Board of the Hail Health Cluster (Approval No. 2024-90), and the study was conducted in accordance with the principles of the Declaration of Helsinki. Informed consent was not required as the study involved no direct contact with patients and used only de-identified data extracted from medical records. Patient confidentiality and data privacy were strictly maintained throughout the research.

### 2.2. HAI Definitions

Healthcare-associated infections (HAIs) were defined according to the Centers for Disease Control and Prevention/National Healthcare Safety Network (CDC/NHSN) criteria [1] as infections that occur ≥48 h after hospital admission, excluding those present or incubating at the time of admission. For the purposes of this study, only HAIs meeting these criteria and developing during ICU stays were included in the analysis. The following device-associated HAIs were identified.

#### 2.2.1. Catheter-Associated Urinary Tract Infection (CAUTI)

A CAUTI was defined as a urinary tract infection in a patient who had an indwelling urinary catheter in place for >2 calendar days on the date of event (day of device placement = Day 1), with the catheter in place on the date of event or the day before, and who met one or more UTI criteria according to CDC/NHSN definitions.

#### 2.2.2. Ventilator-Associated Pneumonia (VAP)

VAP was defined as pneumonia in a patient who had a mechanical ventilator in place for >2 calendar days on the date of event (day of device placement = Day 1), with the ventilator in place on the date of event or the day before, and who met radiologic and clinical criteria according to CDC/NHSN definitions.

#### 2.2.3. Central Line-Associated Bloodstream Infection (CLABSI)

A CLABSI was defined as a primary bloodstream infection in a patient who had a central line in place for >2 calendar days on the date of event (day of device placement = Day 1), with the central line in place on the date of event or the day before, and with no other identified source for the infection.

### 2.3. Microbiological Data Classification

For each HAI episode, microbiological data were reviewed to identify the causative pathogen(s) according to the U.S. Centers for Disease Control and Prevention/National Healthcare Safety Network (CDC/NHSN) surveillance definitions for device-associated infections (CAUTIs, VAP, CLABSIs) [1]. Only organisms isolated from the specimen type specified in the relevant case definition (e.g., urine for CAUTI, respiratory specimen for VAP, blood for CLABSI) and meeting the diagnostic criteria were classified as causative pathogens and included in the main analysis. No single HAI episode in this study had more than one causative organism according to these criteria. In patient-level polymicrobial cases, additional organisms were isolated from other specimen types during the same clinical episode but were not considered causative for the HAI; these co-isolates are documented separately in Appendix A, which also specifies the specimen type, associated device, and polymicrobial status for each record. Because all patient identifiers were removed, it was not possible to determine whether the same patient experienced multiple distinct HAI types during a single hospitalization. This approach ensured that the microbiological profiles reported are directly linked to device-associated HAIs as defined by standardized CDC/NHSN surveillance criteria.

### 2.4. Data Collection

Data were collected retrospectively from hospital records, including patient demographics, medical history, infection details, and invasive device use. Common risk factors for HAIs, such as hypertension, diabetes mellitus, chronic kidney disease, and cerebrovascular accidents, were documented. As this was a cross-sectional study, the sample size was based on the inclusion of all eligible patients identified during the study period. To maintain data quality, records were reviewed for completeness and duplicate entries were removed.

### 2.5. Statistical Analysis

Data were analyzed using the Statistical Package for the Social Sciences (SPSS), version 31 (IBM Corp., Armonk, NY, USA). Descriptive statistics were used to summarize patient demographics, clinical characteristics, and microbiological findings. Categorical variables were reported as frequencies and percentages, while continuous variables were described using means and ranges. Odds ratios (ORs) with 95% confidence intervals (CIs) were calculated to assess the strength of associations. A *p*-value <0.05 was considered statistically significant.

### 2.6. Data Sources and Limitations

All data were extracted from hospital medical records and microbiology laboratory databases. While detailed patient demographics, clinical characteristics, microbiological results, and device use were available, device-day data (e.g., catheter-days, ventilator-days, central line-days) were not recorded in the available datasets. This precluded the calculation of device-associated infection rates per 1000 device-days, which is a standard metric recommended by the CDC/NHSN for benchmarking. In addition, antimicrobial susceptibility testing results and resistance profiles for the isolated pathogens were not consistently documented in the available records. This prevented a detailed analysis of resistance patterns, which is critical for informing infection control strategies and guiding empirical therapy. Both the absence of device-day data and the lack of comprehensive resistance data are acknowledged as methodological limitations and have been addressed in the Discussion section with recommendations for the prospective collection of these metrics in future research.

## 3. Results

### 3.1. Prevalence of Healthcare-Associated Infections

A total of 260 patients admitted to the ICU of a tertiary hospital in Aljouf, Saudi Arabia, between 2020 and 2023 were included in this retrospective study. Admission rate over the 4-year period was highest in 2020 (*n* = 85; 32.6%) and lowest in 2022 (*n* = 39; 15%). This cohort comprised more males (*n* = 147, 56.5%) than females (*n* = 113, 43.5%) with a mean age of 58 years (range 1–118 years). Patients were stratified into 5 age groups with those older than 56 years representing the majority (*n* = 142; 54.6%). Other age groups included ≤25 years (*n* = 27; 10.4%), 26–35 years (*n* = 31; 11.9%), 36–45 years (*n* = 32; 12.3%), and 46–55 years (*n* = 28; 10.8%). Among all ICU patients, 40 (15.4%) were diagnosed with at least one HAI during their stay. The year 2020 accounted for 50% of all HAI cases (*n* = 20), which was followed by 2023 (*n* = 9), 2021 (*n* = 7), and 2022 (*n* = 4). Although the number of HAI cases appeared to decrease over time, the differences in infection rates across the years were not statistically significant. Analysis of HAI occurrence by gender revealed a higher prevalence in males (*n* = 25, 62.5%) than females (*n* = 15, 37.5%); however, this difference was not statistically significant (OR = 1.34, *p* = 0.409). Similarly, patients aged over 56 years had the highest number of HAI cases (*n* = 22); however, no statistically significant association was found between age group and HAI occurrence (all *p* > 0.05) (Table 1).

### 3.2. Microbiological Profile

Across all ICU admissions during the study period, 22 different organisms were isolated from the patients, totaling 307 microbiological findings. Of these, only 11 were associated with confirmed HAIs. The most common organism identified was *K. pneumoniae* (*n* = 14; 35.0%), which was followed by *Acinetobacter baumannii* (*n* = 11; 27.5%) and *Pseudomonas* spp. (*n* = 4; 10.0%) (Table 2). Other isolated pathogens included *Candida albicans* (5.0%), *Candida tropicalis* (2.5%), *Providencia stuartii* (2.5%), and *S. aureus* (2.5%). One HAI case was diagnosed as an infection-related ventilator-associated complication with no growth on culture but meeting clinical diagnostic criteria (for example, elevated temperature, abnormal white blood cell count, and response to antibiotics).

### 3.3. Temporal Trends

Figure 1 illustrates the annual detection frequency of each organism associated with HAIs. In 2020, *K. pneumoniae* and *A. baumannii* dominated, accounting for the highest organism detection frequencies across all years (*K. pneumoniae*, *n* = 8; *A. baumannii*, *n* = 6). Multiple pathogens, including *Pseudomonas aeruginosa*, *Escherichia coli*, and *C. albicans*, were also detected in smaller numbers during that year. In 2021, fewer infections were reported with only *K. pneumoniae* and *A. baumannii* isolated (*n* = 3 each). The years 2022 and 2023 showed greater microbial diversity with multiple organisms such as *C. tropicalis*, *P. stuartii*, *Pseudomonas* spp., and *C. auris* identified at lower but notable frequencies. This temporal distribution highlights a shift from high-frequency Gram-negative bacteria in 2020 to a broader range of organisms, including fungal pathogens, in later years, reflecting potential changes in infection control dynamics or antibiotic usage patterns over time.

### 3.4. Risk Factors Associated with HAI Cases

Comorbid conditions were common among patients with HAIs, with most of them having at least one pre-existing medical condition (*n* = 34, 85%), while the rest had none (*n* = 6, 15%). Cardiovascular diseases were the most prevalent conditions, particularly hypertension (*n* = 12, 30.0%) and ischemic heart disease (*n* = 3, 7.5%). Metabolic conditions (diabetes [30%] and hypothyroidism [7.5%]) and renal disorders (chronic kidney disease [10%] and renal impairment [7.5%]) were also notable (Table 3).

Among the 260 ICU patients, all 40 HAI cases occurred in those who required invasive devices. The most common device combination was the concurrent use of a Foley catheter, central line, and ventilator, accounting for 80.0% (*n* = 32) of all HAIs. An additional infection rate of 7.5% (*n* = 3) was observed in patients who required all four devices, including a tracheal tube. Combinations involving two devices, such as a Foley catheter with either a central line or ventilator (2.5% each), or a central line with a ventilator (5.0%), were observed less frequently. Only one HAI case (2.5%) occurred with single device use (Foley catheter), and no infections were observed in patients who required only a ventilator, only a central line, or no device (Table 4).

Foley catheters were the most frequently associated device, implicated in 47.5% (*n* = 19) of HAI cases, primarily catheter-associated urinary tract infections (CAUTIs). Ventilator use accounted for 35.0% (*n* = 14) of cases, mainly ventilator-associated pneumonia (VAP), while central line use was associated with 17.5% (*n* = 7) of cases, primarily central line-associated bloodstream infections (CLABSIs) (Table 5). *Klebsiella pneumoniae* was the most common pathogen overall, which was responsible for 36.8% of CAUTIs, 21.4% of VAP cases, and 57.1% of CLABSIs. *Acinetobacter baumannii* was the second most frequent isolate, predominating in VAP cases (42.9%) and present in both CAUTIs (10.5%) and CLABSIs (42.9%). Other identified pathogens included *Pseudomonas* spp., multiple *Candida* species (*C. albicans, C. auris, C. tropicalis*), *Staphylococcus aureus*, and less common Gram-negative bacilli such as *Providencia stuartii*, *Serratia marcescens*, and *Proteus mirabilis*.

No individual HAI episode in this study had more than one causative organism as defined by CDC/NHSN criteria. However, 9 out of 40 cases (22.5%) were polymicrobial at the patient level, meaning additional organisms were isolated from other specimen types during the same infection episode but were not classified as causative for the HAI reported in Table 5. These co-isolates are documented in Appendix A, which also provides the specimen type, associated device, and polymicrobial status for each record. Because patient identifiers were de-identified in the dataset, it was not possible to determine whether the same patient experienced multiple distinct HAI types during a single hospitalization.

When assessing the number of invasive devices used, a significant increase in infection risk was observed with higher device counts. The use of three devices was observed in 32 HAI cases, corresponding to an OR of 7.822 (95% CI: 1.031–59.346; *p* = 0.047), indicating nearly eight times greater odds of infection compared with the use of zero or one device. The use of four devices was associated with the highest risk (OR: 14.143, 95% CI: 1.276–156.782; *p* = 0.031), reflecting over 14 times greater odds of acquiring an HAI compared with the reference group (Table 6). Although an increased OR of 2.933 was associated with the use of two devices, this finding did not reach statistical significance (*p* = 0.346).

## 4. Discussion

This retrospective study revealed an HAI prevalence of 15.38% among ICU patients in Aljouf between 2020 and 2023, which was consistent with the WHO average estimate of 15% in developing countries [2]. The global burden of HAIs remains substantial with an estimated prevalence of 7% in developed countries and higher rates in low- and middle-income regions, particularly in critical care units [3]. For instance, Europe records up to 80,000 new HAI cases daily [14], whereas ICUs in Africa experience an endemic HAI prevalence of approximately 50% [15]. These data highlight the persistent challenges posed by HAIs to patient safety worldwide.

The temporal trend observed in the present study showed that half of all HAI cases occurred in 2020, coinciding with the onset of the COVID-19 pandemic. Similar findings were reported by Al-Tawfiq et al. [16], who identified increased HAI rates linked to pandemic-related disruptions. The pandemic likely contributed to prolonged hospital stays, an increased number of invasive procedures, and resource strain, all of which amplify infection risk [8]. Peconi et al. [17]. documented an 11.1% increase in MDRO infections during COVID-19, supporting the association between pandemic conditions and elevated HAI incidence. Thus, the surge in infection rate in 2020 likely reflects both direct and indirect effects of the pandemic on healthcare delivery.

While no statistically significant association was observed between age or gender and HAI cases in the present study, patients aged over 56 years and males comprised the majority of cases, which was consistent with previous findings [18]. Older patients often have diminished immune defenses and multiple comorbidities that increase their vulnerability to infections [19]. Similarly, gender differences may be related to behavioral, physiological, or healthcare exposure variables, although the underlying mechanisms require further investigations. The predominance of HAI in older males highlights the need for targeted infection prevention in these groups.

Microbiological profiling identified *Klebsiella pneumoniae* as the most common pathogen among healthcare-associated infection (HAI) cases (35%), which was followed by *Acinetobacter baumannii* and *Pseudomonas* species. These findings align with previous reports from the region; for example, Al-Khalidi et al. [12] found *Klebsiella pneumoniae* to be the predominant organism in surgical site infections (SSIs) in Aljouf, while methicillin-resistant *Staphylococcus aureus* was the leading pathogen. In this paper, the refined classification of causative agents according to U.S. Centers for Disease Control and Prevention/National Healthcare Safety Network (CDC/NHSN) criteria and the exclusion of non-qualifying co-isolates revealed that *Klebsiella pneumoniae* accounted for 57.1% of central line-associated bloodstream infections (CLABSIs), 36.8% of catheter-associated urinary tract infections (CAUTIs), and 21.4% of ventilator-associated pneumonia (VAP) cases (Table 5). *Acinetobacter baumannii* was the second most frequent isolate, predominating in both CLABSIs and VAP cases (42.9% each) and also present in CAUTIs (10.5%). Other identified pathogens included *Pseudomonas* species, multiple *Candida* species (*Candida albicans*, *Candida auris*, *Candida tropicalis*), *Staphylococcus aureus*, and less common Gram-negative bacilli such as *Providencia stuartii*, *Serratia marcescens*, and *Proteus mirabilis*. The prominence of Gram-negative bacilli, particularly *Klebsiella pneumoniae* and *Acinetobacter baumannii*, is of concern given their propensity for multidrug resistance and association with worse clinical outcomes [6]. The emergence of fungal pathogens, although less frequent, also warrants caution owing to increasing antifungal resistance globally [4]. The predominance of multidrug-resistant organisms (MDROs) such as *Klebsiella pneumoniae* and *Acinetobacter baumannii* in HAI cases highlights the urgent need for antimicrobial stewardship programs and enhanced infection control measures [6]. These pathogens contribute to increased morbidity, mortality, and healthcare costs, posing substantial treatment challenges. The high frequency of device-associated infections by these MDROs indicates that intensive care units (ICUs) should prioritize interventions, including strict hand hygiene, environmental cleaning, and appropriate device use, to limit pathogen transmission [20].

A critical risk factor highlighted in the present study was the use of invasive devices. All HAI cases were associated with the use of one or more invasive devices, particularly combinations of Foley catheters, central lines, and ventilators. The use of three or more devices significantly increased the odds of infection (OR = 14.14), emphasizing the cumulative risk of multiple device insertions [7]. Foley catheters were implicated in nearly half of infections, primarily CAUTIs, mirroring the findings of Alsheddi et al. [9], who reported high urinary catheter utilization ratios in Saudi ICUs. These results reinforce the fact that device-associated infections constitute a major proportion of HAI cases and highlight the importance of rigorous device management protocols.

Furthermore, comorbid conditions were prevalent in HAI cases in our study, mainly hypertension and diabetes mellitus (30% in each case). These chronic illnesses contribute to immune dysfunction, delayed healing, and prolonged hospitalization, thereby increasing HAI susceptibility [21]. Rodríguez-Acelas et al. [19] identified diabetes as a significant risk factor for HAIs, linking metabolic dysregulation to vulnerability. The high burden of comorbidities among patients with HAI in Aljouf suggests that comprehensive patient management addressing underlying diseases is essential for reducing infection risk.

Our findings also highlight the critical role of infection prevention and control (IPC) programs tailored to local epidemiology. For instance, the reduced SSI rates observed in Aljouf hospitals has been attributed to strict infection control measures and active follow-up [12]. Consistent with this, HAI surveillance systems, such as the Healthcare-associated Infection Surveillance Network (HESN) in Saudi Arabia, can provide valuable real-time data to guide IPC strategies [9]. However, data on the effectiveness of such systems in northern regions, such as Aljouf, remain scarce, calling for further evaluation.

Despite its strengths, this study has some limitations that should be acknowledged. The single-center retrospective design and relatively small sample size may limit the generalizability of the findings and introduce potential bias. Larger multicenter studies encompassing diverse healthcare settings would provide a more comprehensive understanding of the epidemiology and risk factors for HAIs in the region. Moreover, the study period included the COVID-19 pandemic, which is a confounding factor that may have influenced infection rates and healthcare practices [8]. Future research should be aimed at disentangling these effects and assessing long-term post-pandemic trends. The COVID-19 pandemic has also highlighted vulnerabilities in healthcare systems, including increased HAI risk owing to staff shortages, increased patient loads, and the disruption of routine IPC protocols [8]. Quigley et al. [22] reported that healthcare workers face a three times greater risk of infection during a pandemic, reflecting broader systemic challenges.

Another methodological limitation was the absence of device-day data (e.g., catheter-days, ventilator-days, central line-days), which precluded the calculation of standardized device-associated infection rates per 1000 device-days—an important benchmark for inter-hospital comparisons and infection control surveillance, as recommended by the CDC/NHSN. Additionally, antimicrobial susceptibility profiles and resistance mechanism data for the isolated microorganisms were unavailable, as these were not consistently recorded in the hospital’s electronic records. Future studies should prospectively collect both device-day and antimicrobial resistance data to inform empiric therapy, strengthen infection control programs, and provide a more comprehensive understanding of the microbiological and clinical implications of HAIs in ICU settings. Addressing these limitations will require investment in healthcare infrastructure, staff training, and contingency planning to maintain effective IPC measures during both routine operations and public health crises.

## 5. Conclusions

In conclusion, this study revealed a substantial burden of HAIs among ICU patients in Aljouf with significant associations with invasive device use and comorbidities such as hypertension and diabetes. *Klebsiella pneumoniae* remains the leading pathogen, emphasizing the threat posed by MDROs. These findings highlight the urgent need to strengthen antimicrobial stewardship, enforce device management protocols, and enhance IPC programs, particularly in high-risk patient populations. Continued surveillance, research, and resource allocation are essential to reduce HAIs and improve patient outcomes in Saudi Arabia and similar global settings.

## Figures and Tables

**Figure 1 microorganisms-13-01916-f001:**
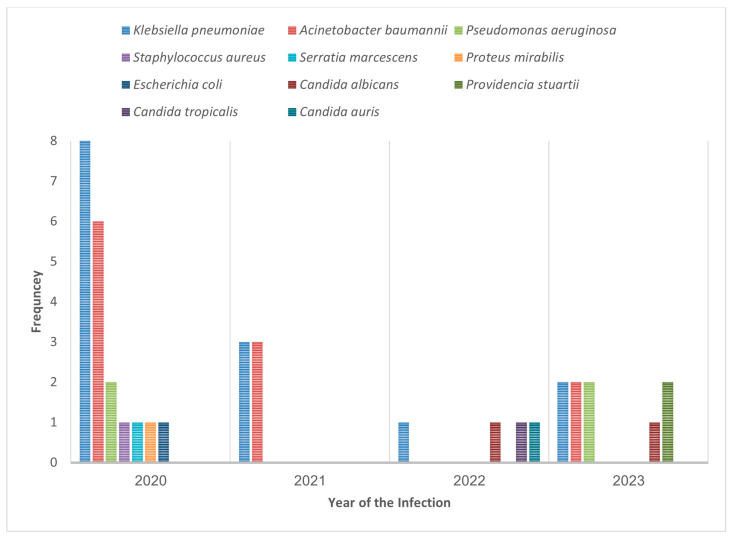
Annual detection frequency of organisms isolated from healthcare-associated infections (2020–2023). The figure presents the distribution of various organisms isolated from healthcare-associated infections over a 4-year period. The x-axis represents the calendar year (2020–2023), while the y-axis indicates the number of isolates identified for each organism per year.

**Table 1 microorganisms-13-01916-t001:** Demographic characteristics and association with HAIs among ICU patients (*n* = 260).

Variable	ICU Patients *n* (%)	HAI Cases *n* (%)	Non-Infected *n* (%)	Odds Ratio	*p*-Value	95% CI
Year of Admission						
2020 (Reference)	85 (32.60)	20 (7.69)	65 (25.00)	1.00	—	—
2021	63 (24.23)	7 (2.69)	56 (21.54)	0.406	0.058	0.160–1.032
2022	39 (15.00)	4 (1.54)	35 (13.46)	0.371	0.091	0.118–1.172
2023	73 (28.08)	9 (3.46)	64 (24.62)	0.457	0.074	0.194–1.079
Gender						
Male (Reference)	147 (56.54)	25 (9.62)	122 (46.92)	1.00	—	—
Female	113 (43.46)	15 (5.77)	98 (37.69)	1.339	0.409	0.669–2.678
Age Group (years)						
Mean (Range)	58 (1–118)	—	—	—	—	—
≤25 (Reference)	27 (10.38)	3 (1.15)	24 (9.23)	1.00	—	—
26–35	31 (11.92)	4 (1.54)	27 (10.38)	1.185	0.835	0.241–5.839
36–45	32 (12.31)	6 (2.31)	26 (10.00)	1.846	0.421	0.415–8.215
46–55	28 (10.77)	5 (1.92)	23 (8.85)	1.739	0.482	0.372–8.124
>56	142 (54.62)	22 (8.46)	120 (46.15)	1.467	0.559	0.406–5.293
Total	260 (100.0)	40 (15.38)	220 (84.62)	—	—	—

**Table 2 microorganisms-13-01916-t002:** Distribution of organisms in ICU and HAI patients (2020–2023).

Organism	All ICU Patients (*n* = 260) No. (%)	HAI Patients (*n* = 40) No. (%)
*Acinetobacter baumannii*	87 (28.34)	11 (27.5)
*Klebsiella pneumoniae*	74 (24.10)	14 (35.0)
*Pseudomonas* spp.	38 (12.38)	4 (10.0)
*Proteus mirabilis*	26 (8.47)	1 (2.5)
*Escherichia coli*	24 (7.82)	1 (2.5)
*Providencia stuartii*	10 (3.26)	2 (5.0)
*Staphylococcus aureus*	8 (2.61)	1 (2.5)
*Candida albicans*	6 (1.95)	2 (5.0)
*Candida auris*	6 (1.95)	1 (2.5)
*Serratia marcescens*	6 (1.95)	1 (2.5)
*Stenotrophomonas maltophilia*	4 (1.30)	—
*Candida tropicalis*	3 (0.98)	1 (2.5)
MRSA (Methicillin-resistant *Staphylococcus aureus*)	3 (0.98)	—
*Klebsiella aerogenes*	3 (0.98)	—
*Enterobacter cloacae*	2 (0.65)	—
*Enterococcus* spp.	2 (0.65)	—
*Citrobacter freundii*	1 (0.33)	—
*Klebsiella pneumoniae* subsp. *ozaenae*	1 (0.33)	—
*Providencia rettgeri*	1 (0.33)	—
*Salmonella* spp.	1 (0.33)	—
*Staphylococcus epidermidis*	1 (0.33)	—
No growth (IVAC) *	—	1 (2.5)
Total	307 (100)	40 (100)

* IVAC: infection-related ventilator-associated complications. The presence of a specific organism is not required to confirm an HAI; the diagnosis is based on abnormal white blood cell (WBC) counts, temperature irregularities, and antibiotic administration patterns.

**Table 3 microorganisms-13-01916-t003:** Comprehensive risk factor profile among patients with healthcare-associated infections (*n*= 40).

Category	Specific Risk Factor	*n*	Percentage of Patients
Cardiovascular (40.0%)	Hypertension (HTN)	12	30.0%
	Ischemic Heart Disease (IHD)	3	7.5%
	Ventricular Septal Defect (VSD)	2	5.0%
	Cardiomyopathy	1	2.5%
	Intracerebral Hemorrhage (IVH)	1	2.5%
Metabolic (35.0%)	Diabetes Mellitus (DM)	12	30.0%
	Hypothyroidism	3	7.5%
Renal (22.5%)	Chronic Kidney Disease (CKD)	4	10.0%
	Renal Impairment	3	7.5%
	End-Stage Renal Disease (ESRD)	2	5.0%
	Nephropathy	1	2.5%
Neurological (10.0%)	Cerebrovascular Accident (CVA)	1	2.5%
	Dementia	1	2.5%
	Hydrocephalus	1	2.5%
	Subdural Hematoma (SDH)	1	2.5%
Gastrointestinal (7.5%)	Peptic Ulcer Disease (PUD)	2	5.0%
	Liver Cirrhosis	1	2.5%
Pulmonary (5.0%)	Chronic Obstructive Pulmonary Disease (COPD)	2	5.0%
Other (17.5%)	Road Traffic Accident (RTA)	6	15.0%
	Bedsore	1	2.5%
Risk Factor Count per Patient	No Documented Risk Factor	6	15.0%
	1 Risk Factor	17	42.5%
	2 Risk Factors	11	27.5%
	≥3 Risk Factors	6	15.0%

**Table 4 microorganisms-13-01916-t004:** Distribution of invasive device combinations and healthcare-associated infections (HAIs) among ICU patients (*n* = 260).

Number of Devices	Device Combination	Patients Admitted *n* (%)	Non-Infected *n* (%)	HAI Cases *n* (%)	HAI Cases to Total HAI (*n* = 40) *n* (%)
4	Foley Catheter + Central Line + Ventilator + Tracheal Tube	10 (3.8%)	7 (2.7%)	3 (1.2%)	3 (7.5%)
3	Foley Catheter + Central Line + Ventilator	165 (63.5%)	133 (51.2%)	32 (12.3%)	32 (80.0%)
3	Foley Catheter + Ventilator + Tracheal Tube	2 (0.8%)	2 (0.8%)	0 (0.0%)	0 (0.0%)
2	Foley Catheter + Central Line	18 (6.9%)	17 (6.5%)	1 (0.4%)	1 (2.5%)
2	Foley Catheter + Ventilator	17 (6.5%)	16 (6.2%)	1 (0.4%)	1 (2.5%)
2	Central Line + Ventilator	14 (5.4%)	12 (4.6%)	2 (0.8%)	2 (5.0%)
1	Foley Catheter	23 (8.8%)	22 (8.5%)	1 (0.4%)	1 (2.5%)
1	Ventilator	3 (1.2%)	3 (1.2%)	0 (0.0%)	0 (0.0%)
1	Central Line	1 (0.4%)	1 (0.4%)	0 (0.0%)	0 (0.0%)
0	No Device Used	7 (2.7%)	7 (2.7%)	0 (0.0%)	0 (0.0%)
	Total	260(100%)	220 (84.6%)	40 (15.4%)	40 (100%)

**Table 5 microorganisms-13-01916-t005:** Device association and causative agents of healthcare-associated infections (*n* = 40).

HAI Type	Associated Device	Episodes *n* (%)	Causative Agent *	*n* (%)
**CAUTI**	Foley Catheter	19 (47.5)	*Klebsiella pneumoniae*	7 (36.8)
			*Candida albicans*	2 (10.5)
			*Acinetobacter baumannii*	2 (10.5)
			*Pseudomonas* spp.	2 (10.5)
			*Staphylococcus aureus*	1 (5.3)
			*Providencia stuartii*	1 (5.3)
			*Candida auris*	1 (5.3)
			*Escherichia coli*	1 (5.3)
			*Candida tropicalis*	1 (5.3)
			*Serratia marcescens*	1 (5.3)
**CLABSI**	Central Line	7 (17.5)	*Klebsiella pneumoniae*	4 (57.1)
			*Acinetobacter baumannii*	3 (42.9)
**VAP**	Ventilator	14 (35)	*Acinetobacter baumannii*	6 (42.9)
			*Klebsiella pneumoniae*	3 (21.4)
			*Pseudomonas* spp.	2 (14.3)
			*Providencia stuartii*	1 (7.1)
			*Proteus mirabilis*	1 (7.1)
			Nil growth (IVAC)	1 (7.1)

***** Causative agents are defined as organisms meeting CDC/NHSN case definitions for CAUTIs, CLABSIs, and VAP. One ventilator-associated event classified as IVAC (nil growth) is included for completeness. Percentages for organisms are calculated relative to the total episodes for each HAI type. **Abbreviations:** HAI, healthcare-associated infection; CAUTI, catheter-associated urinary tract infection; VAP, ventilator-associated pneumonia; CLABSI, central line-associated bloodstream infection; IVAC, infection-related ventilator-associated complication.

**Table 6 microorganisms-13-01916-t006:** Association between number of invasive devices used and risk of healthcare-associated infections (*n* = 260).

Device Use Category	ICU Patients *n* (%)	HAI Cases *n* (%)	Non-Infected *n* (%)	Odds Ratio	*p*-Value	95% CI
0–1 device (Reference)	34 (13.08)	1 (0.38)	33 (12.69)	1.00	—	—
2 devices	49 (18.85)	4 (1.54)	45 (17.31)	2.933	0.346	0.313–27.468
3 devices	167 (64.23)	32 (12.31)	135 (51.92)	7.822	0.047 *	1.031–59.346
4 devices	10 (3.85)	3 (1.15)	7 (2.69)	14.143	0.031 *	1.276–156.782
Total	260 (100.0)	40 (15.38)	220 (84.62)	—	—	—

* Significant at 0.05.

## Data Availability

The data supporting the findings of this study are available from the corresponding author upon reasonable request. Due to privacy and ethical restrictions related to patient information, the data are not publicly available.

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
