# Peer review of "Prevalence, Microbiological Profile, and Risk Factors of Healthcare-Associated Infections in Intensive Care Units: A Retrospective Study in Aljouf, Saudi Arabia"

_microorganisms, 2025, doi:10.3390/microorganisms13081916_

Round 1
Reviewer 1 Report
Comments and Suggestions for Authors
- This study is a retrospective cross-sectional design. To use the term "prevalence" was more suitable in descriptive epidemiology results.
- The authors should state the definitions of each HAI and present the data.
- The authors should better present the relationship between microorganisms and different HAI items.
- Did the patients have different HAIs and multiple microorganisms in the same hospitalized course? These data should be collected and presented.
- Incidence rates and the usage of device-day should also be calculated and presented. The manuscript only presented the numbers of device usage and the relationship between HAIs.
Reviewer 2 Report
Comments and Suggestions for Authors
The issue of hospital infections is of interest to microbiologists, epidemiologists, and physicians around the world. Despite knowledge of risk factors, mechanisms contributing to their occurrence, and methods of reducing them, their number is not decreasing. Recognizing the scale and etiology of hospital infections is very important for developing and implementing appropriate measures and procedures to reduce their occurrence and eliminate the spread of existing ones.
In many, if not most, countries, there are no central systems for monitoring the incidence of hospital-acquired infections, and therefore, knowledge about them is still limited. Numerous studies aimed at assessing the scale of hospital-acquired infections undertaken on a national or international scale provide only approximate, albeit relevant, data on this phenomenon.
The issue of hospital infections in many regions of the world is still not sufficiently understood, which is why the manuscript by Alshammari and Alruwaili is a good addition to the knowledge on these infections.
The work submitted for review is generally good, although I find several issues that require correction or clarification.
Why are young adult patients (aged >=18-25) and children (aged 0-<18) included in a single age group up to 25 years of age?
What was the number of patients under 18 years of age?
If the number of patients under 18 was small, I suggest stating this in the text as a justification for combining pediatric patients with adults.
What does “(Ref)” mean in Table 1, in the Variable column?
In Table 2, Klebsiella aerogenes and Enterobacter aerogenes are the same organism. The current name is Klebsiella aerogenes.
Klebsiella ozaenae proper name is Klebsiella pneumoniae subsp. ozaenae.
A significant shortcoming of the study is the lack of information on the drug sensitivity of the isolated microorganisms and the detected mechanisms of drug resistance. In addition, the study would be more valuable if it described the microbiological screening methods, if used, to assess patient colonization upon admission to the ICU. Also, the manuscript lacks a clear definition of whether the HAIs described are those that developed during hospitalization or whether HAIs with which the patient was admitted to the ICU are also included.
Furthermore, I believe that to present the epidemiology of HAIs better, the manuscript should include an analysis of the incidence of HAIs, taking into account the total period of hospitalization (before admission to the ICU) and the period of hospitalization in the ICU.
Round 2
Reviewer 1 Report
Comments and Suggestions for Authors
The current version of the manuscript is much better than the original version, providing more appropriate information to improve the understanding of the study. I agree to accept the current manuscript for publication.